# NON-EXCHANGEABLE CONFORMAL RISK CONTROL

**António Farinhas** [1,2], **Chrysoula Zerva** [1,2], **Dennis Ulmer** [3,4], **André F. T. Martins** [1,2,5]

[1]Instituto de Telecomunicações,
[2]Instituto Superior Técnico, Universidade de Lisboa (Lisbon ELLIS Unit),
[3]IT University of Copenhagen,  [4]Pioneer Centre for Artificial Intelligence ,  [5]Unbabel
`{antonio.farinhas,chrysoula.zerva,andre.t.martins}@tecnico.ulisboa.pt,`
`dennis.ulmer@mailbox.org`

## ABSTRACT

Split conformal prediction has recently sparked great interest due to its ability to provide formally guaranteed uncertainty sets or intervals for predictions made by black-box neural models, ensuring a predefined probability of containing the actual ground truth. While the original formulation assumes data exchangeability, some extensions handle non-exchangeable data, which is often the case in many real-world scenarios. In parallel, some progress has been made in conformal methods that provide statistical guarantees for a broader range of objectives, such as bounding the best $F_1$-score or minimizing the false negative rate in expectation. In this paper, we leverage and extend these two lines of work by proposing *non-exchangeable conformal risk control*, which allows controlling the expected value of any monotone loss function when the data is not exchangeable. Our framework is flexible, makes very few assumptions, and allows weighting the data based on its relevance for a given test example; a careful choice of weights may result in tighter bounds, making our framework useful in the presence of change points, time series, or other forms of distribution drift. Experiments with both synthetic and real world data show the usefulness of our method.

## 1 INTRODUCTION

As the use of machine learning systems for automated decision-making becomes more widespread, the demand for these systems to produce reliable and trustworthy predictions has grown significantly. In this context, conformal prediction (Papadopoulos et al., 2002; Vovk et al., 2005) has recently resurfaced as an attractive framework. Instead of providing a single output, this framework creates prediction sets or intervals that inherently account for uncertainty. These sets come with a statistical guarantee known as **coverage**, which ensures that they contain the ground truth in expectation, thereby providing a formal promise of reliability.

The standard formulation of conformal prediction has, however, important limitations. First, it assumes that all data is **exchangeable**, a condition which is often violated in practice (*e.g.*, when there is correlation over time or space). Second, while the predicted sets/intervals provide guarantees on coverage, they do not bound arbitrary losses, some of which may be more relevant for the situation at hand (*e.g.*, the $F_1$-score or the false negative rate in multilabel classification problems). Several works have been proposed to improve over these two shortcomings, namely through non-exchangeable conformal prediction (Tibshirani et al., 2019; Gibbs & Candes, 2021; Barber et al., 2023) and conformal risk control (Bates et al., 2021; Angelopoulos et al., 2023a, CRC). In this paper, we extend these lines of research and propose **non-exchangeable conformal risk control** (non-X CRC). Our main contributions are:

• We propose a new method for conformal risk control that provides formal guarantees when the data is not exchangeable, while also achieving the same guarantees as existing methods if the data is in fact exchangeable (see Table 1 where we position our work in the literature);

• Theorem 1 establishes a new bound on the expected loss (assumed to be monotonic and bounded), allowing weighting the calibration data based on its relevance for a given test example;

Table 1: Our framework combines two approaches, non-exchangeable conformal prediction and conformal risk control. Through this combination we are able to control the expected value of arbitrary monotonic loss functions when the data is not exchangeable, extending both frameworks.

| Method | Data assumptions | Loss |
|---|---|---|
| Papadopoulos et al. (2002) | exchangeable | miscoverage |
| Barber et al. (2023) | ✗ | miscoverage |
| Angelopoulos et al. (2023a) | exchangeable | nonincreasing, arbitrary |
| Angelopoulos et al. (2023a, Prop. 3) | covariate shift, known likelihood ratio | nonincreasing, arbitrary |
| **This paper** | ✗ | nonincreasing, arbitrary |

- We demonstrate the usefulness of our framework on three tasks: multilabel classification on synthetic data by minimizing the false negative rate; monitoring electricity usage by minimizing the $\lambda$-insensitive absolute loss; and open-domain question answering by bounding the best $F_1$-score.[1]

Throughout the paper, we use the following definition of exchangeable data distribution, which is a weaker assumption than independent and identically distributed (i.i.d.) data.

**Definition 1** (**Exchangeable data distribution**). *Let $\mathcal{X}$ and $\mathcal{Y}$ designate input and output spaces. A data distribution in $\mathcal{X} \times \mathcal{Y}$ is said to be exchangeable if and only if we have $\mathbb{P}((X_{\pi(1)}, Y_{\pi(1)}), \dots, (X_{\pi(n)}, Y_{\pi(n)})) = \mathbb{P}((X_1, Y_1), \dots, (X_n, Y_n))$ for any finite sample $\{(X_i, Y_i)\}_{i=1}^n \subseteq \mathcal{X} \times \mathcal{Y}$ and any permutation function $\pi$. Note that if the data distribution is i.i.d., then it is also exchangeable, since $\mathbb{P}((X_1, Y_1), \dots, (X_n, Y_n)) = \prod_{i=1}^n \mathbb{P}((X_i, Y_i))$.*

## 2 BACKGROUND

We start by providing background on conformal prediction (Papadopoulos et al., 2002; Vovk et al., 2005) in §2.1. We then discuss recent extensions of the framework—§2.2 discusses the case where the data is non-exchangeable (Barber et al., 2023), which is often the case when models are deployed in practice. Another extension pivots from guaranteeing coverage to instead constraining the expected value of any monotone loss function (Angelopoulos et al., 2023a), useful for tasks in which the natural notion of error is not miscoverage (§2.3).

### 2.1 CONFORMAL PREDICTION

Although other methods exist, this paper focuses on *split* conformal prediction (Papadopoulos et al., 2002; hereinafter referred to simply as *conformal prediction*). We start with a pretrained model and measure its performance on a calibration set $\{(X_i, Y_i)\}_{i=1}^n$ of paired examples. Under the assumption of exchangeable data $\{(X_i, Y_i)\}_{i=1}^{n+1}$, conformal prediction constructs prediction sets with the following coverage guarantee:

$$\mathbb{P}\big(Y_{n+1} \in \mathcal{C}(X_{n+1})\big) \geq 1 - \alpha, \tag{1}$$

where $(X_{n+1}, Y_{n+1})$ is a new data point and $\alpha$ a predefined confidence level. This is accomplished through the following steps: Let $s(x, y) \in \mathbb{R}$ be a non-conformity score function, where larger scores indicate worse agreement between $x$ and $y$. We compute the value $\hat{q}$ as the $1/n\lceil (n+1)(1-\alpha) \rceil$ quantile of the calibration scores and construct a prediction set as follows:

$$\mathcal{C}(X_{n+1}) = \big\{y : s(X_{n+1}, y) \leq \hat{q}\big\}. \tag{2}$$

This prediction set satisfies the coverage guarantee in Eq. (1), see *e.g.*, Angelopoulos & Bates, 2021, App. D for a proof. While this guarantee helps to ensure a certain reliability of the calibrated model, the assumption of exchangeable data is often not true when models are deployed in practice, *e.g.*, due to distribution drift in time series or correlations between different data points.

---

[1]Our code is available at https://github.com/deep-spin/non-exchangeable-crc.

## 2.2 NON-EXCHANGEABLE CONFORMAL PREDICTION

Let us now consider prespecified weights $\{w_i\}_{i=1}^n \in [0,1]^n$ and define $\tilde{w}_i := w_i/(1 + \sum_{i=1}^N w_i)$. We take a look at a generalization of conformal prediction put together by Barber et al. (2023), which provides the following coverage guarantee, also valid when exchangeability is violated:

$$\mathbb{P}\big(Y_{n+1} \in \mathcal{C}(X_{n+1})\big) \geq 1 - \alpha - \sum_{i=1}^n \tilde{w}_i d_{\mathrm{TV}}(Z, Z^i), \tag{3}$$

where $Z := (X_1, Y_1), \ldots, (X_n, Y_n), (X_{n+1}, Y_{n+1})$ is a sequence of $n$ calibration examples followed by a test example, $Z^i$ denotes $Z$ after swapping $(X_i, Y_i)$ with $(X_{n+1}, Y_{n+1})$, and $d_{\mathrm{TV}}(Z, Z^i)$ is the total variation (TV) distance between $Z$ and $Z^i$. This is accomplished by using

$$\hat{q} = \inf \left\{ q : \sum_{i=1}^N \tilde{w}_i \mathbf{1}\big\{s_i \leq q\big\} \geq 1 - \alpha \right\} \tag{4}$$

to construct prediction sets the same way as in Eq. (2). See Barber et al. (2023, §4) for a proof. It is worth noting that this method recovers standard conformal prediction when $\{w_i\}_{i=1}^n = 1$. Besides, if the data is exchangeable, then the distribution of $Z$ is equal to the distribution of $Z^i$, and thus using a weighted procedure does not hurt coverage according to Eq. (3), since $d_{\mathrm{TV}}(Z, Z^i) = 0$ for all $i$. Intuitively, the "closer" to exchangeable the data is, the smaller the last term will be in Eq. (3). By choosing wisely the weights $w_i$—e.g., by setting large weights to calibration points $(x_i, y_i)$ such that $Z$ and $Z^i$ are similarly distributed and smaller weights otherwise—tighter bounds can be obtained. For example, in time series data we may want to place larger weights on more recent observations.

## 2.3 CONFORMAL RISK CONTROL

Let us now consider an additional parameter $\lambda$ and construct prediction sets of the form $\mathcal{C}_\lambda(\cdot)$, where larger $\lambda$ yield larger prediction sets, *i.e.*, $\lambda \leq \lambda' \implies \mathcal{C}_\lambda(.) \subseteq \mathcal{C}_{\lambda'}(.)$ (see Angelopoulos & Bates (2021, §4.3) for an example). Let $\ell$ be an arbitrary (bounded) loss function that shrinks as $\mathcal{C}(X_{n+1})$ grows (*i.e.*, that is **monotonically nonincreasing** with respect to $\lambda$). We switch from conformal methods that provide prediction sets that bound the miscoverage $\mathbb{P}\big(Y_{n+1} \notin \mathcal{C}(X_{n+1})\big) \leq \alpha$ to conformal risk control (Angelopoulos et al., 2023a), which provides guarantees of the form

$$\mathbb{E}\Big[\underbrace{\ell(\mathcal{C}(X_{n+1}), Y_{n+1})}_{L_{n+1}(\hat{\lambda})}\Big] \leq \alpha. \tag{5}$$

This is accomplished as follows. Let $L_i(\lambda) = \ell(\mathcal{C}_\lambda(X_i), Y_i)$, $i = 1, \ldots, n+1$, with $L_i : \Lambda \to (-\infty, B]$ and $\lambda_{\max} := \sup \Lambda$, be an exchangeable collection of nonincreasing functions of $\lambda$. Choosing an optimal $\hat{\lambda}$ as

$$\hat{\lambda} = \inf \left\{ \lambda : \frac{n}{n+1} \hat{R}_n(\lambda) + \frac{B}{n+1} \leq \alpha \right\}, \quad \hat{R}_n(\lambda) = \frac{1}{n} \sum_{i=1}^n L_i(\lambda), \tag{6}$$

yields the guarantee in Eq. (5), see Angelopoulos et al. (2023a, §2) for a proof. When $\ell(\mathcal{C}(X_{n+1}), Y_{n+1}) = \mathbf{1}\big\{Y_{n+1} \notin \mathcal{C}(X_{n+1})\big\}$ is the miscoverage loss, we recover standard conformal prediction (§2.1). Note that, as required, this loss is nonincreasing. Other nonincreasing losses include the false negative rate, $\lambda$-insensitive absolute error, and the best token-level $F_1$-loss, all of which used in our experiments in §4. A limitation of the construction presented in this section is that it relies on the assumption of data exchangeability, which might be violated in practical settings. Our work circumvents this requirement, as we show next.

## 3 NON-EXCHANGEABLE CONFORMAL RISK CONTROL

Up to this point, we have described how to construct prediction sets/intervals with coverage guarantees for non-exchangeable data, in §2.2, and how to control the expected value of arbitrary monotone loss functions, when the data is exchangeable, in §2.3. Using the same notation as before, we now

present our method, *non-exchangeable conformal risk control*, which puts together these parallel lines of research, providing guarantees of the form:

$$\mathbb{E}[L(\hat{\lambda}; (X_{n+1}, Y_{n+1}))] \leq \alpha + (B - A) \sum_{i=1}^{n} \tilde{w}_i d_{\mathrm{TV}}(Z, Z^i), \tag{7}$$

where we additionally assume $A < B < \infty$ to be a lower bound on $L_i : \Lambda \to [A, B]$. Let us define $N_w := \sum_{i=1}^{N} w_i$. Eq. (7) is obtained by choosing an optimal $\hat{\lambda}$ as

$$\hat{\lambda} = \inf \left\{ \lambda : \frac{N_w}{N_w + 1} \hat{R}_n(\lambda) + \frac{B}{N_w + 1} \leq \alpha \right\}, \quad \hat{R}_n(\lambda) = \frac{1}{N_w} \sum_{i=1}^{n} w_i L(\lambda; (x_i, y_i)). \tag{8}$$

We can see how Eq. (7) simultaneously mirrors both Eq. (3) and Eq. (5): for an optimal choice of $\lambda$, the expected risk for a new test point is bounded by $\alpha$ plus an extra loosening term that depends on the normalized weights $\{w_i\}_{i=1}^{n}$ and on the total variation distance between $Z$ and $Z^i$. When the data is in fact exchangeable, we have again $d_{\mathrm{TV}}(Z, Z^i) = 0$ for all $i$, and we recover Eq. (5), *i.e.*, our method achieves the same coverage guarantees as standard conformal risk control. Although our theoretical bound in Eq. (7) holds for any choice of weights, this result is only useful when the loosening term is small, *i.e.*, if we choose small weights $w_i$ for data points $Z^i$ with large total variation distance $d_{\mathrm{TV}}(Z, Z^i)$. While the true value of this term is typically unknown, in some situations, such as distribution drift in time series, we expect it to decrease with $i$, motivating the choice of weights that increase with $i$. The same principle can be applied in other domains (*e.g.*, for spatial data, one may place higher weights to points close in space to the test point). We come back to this point in §3.2.

The result in Eq. (7) is valid when the weights are fixed, *i.e.*, data-independent. However, our result still applies in the case of data-dependent weights $w_i = w(X_i, X_{n+1})$ if we replace $\sum_{i=1}^{n} \tilde{w}_i d_{\mathrm{TV}}(Z, Z^i)$ by $\mathbb{E}\left[\sum_{i=1}^{n} \tilde{w}_i d_{\mathrm{TV}}(Z, Z^i | w_1, \ldots, w_n)\right]$ (see Barber et al. (2023), §4.5) for more information). We experiment with this approach in §4.3, where $w_i$ is a function of the embedding similarity between $X_i$ and $X_{n+1}$, showing that the new bound is still useful in practice.

## 3.1 FORMAL GUARANTEES

Now that we have presented an overview of our method, we proceed to providing a formal proof for the guarantee in Eq. (7). We begin with a lemma, proved in App. A, that establishes a TV bound that extends the one introduced by Barber et al. (2023):

**Lemma 1.** *Let $f : S \to [A, B] \subset \mathbb{R}$ be a bounded function on a measurable space $(S, \mathcal{A})$ (where $\mathcal{A} \subseteq 2^S$ is a $\sigma$-algebra) and let $P$ and $Q$ be two probability measures on $(S, \mathcal{A})$. Then*

$$|\mathbb{E}_P[f] - \mathbb{E}_Q[f]| \leq (B - A)d_{\mathrm{TV}}(P, Q). \tag{9}$$

Note that when $f(t) = \mathbf{1}\{t \in V\}$ for some event $V \in \mathcal{A}$, the left-hand side becomes $|P(V) - Q(V)|$ and we recover the bound used in the proof of Barber et al. (2023, §6.2).

We now state the main result. The proof technique is similar to that of Barber et al. (2023), but instead of modeling the event of a variable belonging to a "strange set", we model expectations of loss functions that depend on a calibration variable. See App. B for the full proof.

**Theorem 1 (Non-exchangeable conformal risk control).** *Assume that for all $(x, y) \in \mathcal{X} \times \mathcal{Y}$ the loss $L(\lambda; (x, y))$ is nonincreasing in $\lambda$ and bounded as $A \leq L(\lambda; (x, y)) \leq B$ for any $\lambda$. Let*

$$Z := (X_1, Y_1), \ldots, (X_n, Y_n), (X_{n+1}, Y_{n+1})$$

*be a sequence of $n$ calibration examples followed by a test example, and let $w_1, \ldots, w_n \in [0, 1]^n$ be data-independent weights. Define $N_w = \sum_{i=1}^{n} w_i$, $\tilde{w}_i = w_i/(N_w + 1)$ for $i \in [n]$ and $\tilde{w}_{n+1} = 1/(N_w + 1)$. Let $\alpha \in [A, B]$ be the maximum tolerable risk, and define*

$$\hat{\lambda} = \inf \left\{ \lambda : \frac{N_w}{N_w + 1} \hat{R}_n(\lambda) + \frac{B}{N_w + 1} \leq \alpha \right\}, \tag{10}$$

*where $\hat{R}_n(\lambda)$ is the weighted empirical risk in the calibration set:*

$$\hat{R}_n(\lambda) = \frac{1}{N_w} \sum_{i=1}^{n} w_i L(\lambda; (x_i, y_i)). \tag{11}$$

*Then, we have*

$$\mathbb{E}[L(\hat{\lambda}; (X_{n+1}, Y_{n+1}))] \leq \alpha + (B - A) \sum_{i=1}^{n} \tilde{w}_i d_{\mathrm{TV}}(Z, Z^i), \tag{12}$$

*where $Z^i$ is obtained from $Z$ by swapping $(X_i, Y_i)$ and $(X_{n+1}, Y_{n+1})$.*

The next section illustrates how we can make practical use of this result to minimize loss functions beyond the miscoverage loss in the presence of non-exchangeable data distributions.

### 3.2 How to choose weights

To make practical use of Theorem 1, we need a procedure to choose the weights $w_i$. We next suggest a strategy based on regularized minimization of the coverage gap $g(\tilde{w}_1, ..., \tilde{w}_n) := (B - A) \sum_{i=1}^{n} \tilde{w}_i d_{\mathrm{TV}}(Z, Z^i)$ via the maximum entropy principle (Jaynes, 1957). Note first that simply minimizing this gap would lead to $\tilde{w}_i = 0$ for all $i \in [n]$ and $\tilde{w}_{n+1} = 1$, which ignores all the calibration data and leads to an infeasible $\hat{\lambda}$ in Eq. (6). In general, if all weights $w_i$ are too small, this leads to a very large $w_{n+1}$ and an unreasonably large $\hat{\lambda}$. On the other extreme, having all weights too large (e.g. $w_i = 1$ for all $i$, which leads to $\tilde{w}_i = 1/(n + 1)$ for $i \in [n + 1]$) ignores the non-exchangeability of the data and may lead to a large coverage gap. Therefore, it is necessary to find a good balance between ensuring a small coverage gap but at the same time ensuring that the distribution $\tilde{w}_1, ..., \tilde{w}_{n+1}$ is not too peaked, i.e., that it has sufficiently high entropy. Since by definition, we must have $\tilde{w}_{n+1} \geq \tilde{w}_i$ for all $i \in [n]$, this can be formalized as the following regularized minimization problem:

$$\min_{\tilde{w}_1, ..., \tilde{w}_{n+1}} (B - A) \sum_{i=1}^{n} \tilde{w}_i d_{\mathrm{TV}}(Z, Z^i) - \beta H(\tilde{w}_1, ..., \tilde{w}_{n+1})$$

$$\text{subject to } \sum_{i=1}^{n+1} \tilde{w}_i = 1 \text{ and } 0 \leq \tilde{w}_i \leq \tilde{w}_{n+1} \text{ for all } i \in [n], \tag{13}$$

where $H(\tilde{w}_1, ..., \tilde{w}_{n+1}) = -\sum_{i=1}^{n+1} \tilde{w}_i \log \tilde{w}_i$ is the entropy function and $\beta > 0$ is a temperature parameter. The solution of this problem is $\tilde{w}_i \propto \exp(-\beta(B - A)d_{\mathrm{TV}}(Z, Z^i))$ for $i \in [n + 1]$.

Although in general $d_{\mathrm{TV}}(Z, Z^i)$ is not known, it is possible in some scenarios to bound or to estimate this quantity: for example, when variables are independent but not identically distributed, it can be shown that $d_{\mathrm{TV}}(Z, Z^i) \leq 2d_{\mathrm{TV}}(Z_i, Z_{n+1})$ (Barber et al., 2023, Lemma 1); and it is possible to upper bound the total variation distance as a function of the (more tractable and amenable to estimation) Kullback-Leibler divergence, e.g., via Pinsker's or Bretagnolle-Huber's inequalities (Bretagnolle & Huber, 1979; Csiszár & Körner, 2011), which may provide good heuristics. For example, in a time series under a distribution shift scenario bounded with a Lipschitz-type condition $d_{\mathrm{TV}}(Z_i, Z_{n+1}) \leq \epsilon(n + 1 - i)$ for some $\epsilon > 0$ (see e.g. (Barber et al., 2023, §4.4)), we could replace $d_{\mathrm{TV}}(Z, Z^i)$ in Eq. (13) by this upper bound to obtain the maxent solution $\tilde{w}_i \propto \exp(-\beta\epsilon(n + 1 - i)) = \rho^{n+1-i}$, where $\rho = \exp(-\beta\epsilon) \in (0, 1)$. This exponential decay of the weights was suggested by (Barber et al., 2023); our maximum entropy heuristic provides further justification for that choice. We use this strategy in some of our experiments in §4.

## 4 Experiments

In this section, we turn to demonstrating the validity of our theoretical results in three different tasks using different nonincreasing losses: a **multilabel classification** problem using synthetic time series data, minimizing the false negative rate (§4.1), a problem involving **monitoring electricity usage**, minimizing the $\lambda$-insensitive absolute loss (§4.2), and an **open-domain question answering** (QA)

task, where we control the best token-level $F_1$-score (§4.3). Throughout, we report our method alongside a conformal risk control (CRC) baseline that predicts $\hat{\lambda}$ following Eq. (6).

## 4.1 MULTILABEL CLASSIFICATION IN A TIME SERIES

We start by validating our approach on synthetic data, before moving to real-world data in the following subsections. To this end, we modified the synthetic regression experiment of Barber et al. (2023, §5.1) to turn it into a multilabel classification problem with up to $M = 10$ different labels. We consider three different setups:

1. **Exchangeable (i.i.d.) data:** We sample $N = 2000$ i.i.d. data points $(X_i, Y_i) \in \mathbb{R}^M \times \mathbb{R}^M$. We sample $X_i$ from a Gaussian distribution, $X_i \overset{\text{iid}}{\sim} \mathcal{N}(\mathbf{0}, \boldsymbol{I}_M)$, and we set $Y_i \sim \mathbf{sign}(\boldsymbol{W} X_i + \boldsymbol{b} + .1\mathcal{N}(\mathbf{0}, \boldsymbol{I}_M))$. The coefficient matrix $\boldsymbol{W}$ is set to the identity matrix $\boldsymbol{I}_M$ and the biases to $\boldsymbol{b} = -\mathbf{0.5}$, to encourage a sparse set of labels.

2. **Changepoints:** We follow setting (1) and sample $N = 2000$ i.i.d. data points $(X_i, Y_i)$, setting $X_i \overset{\text{iid}}{\sim} \mathcal{N}(\mathbf{0}, \boldsymbol{I}_M)$ and $Y_i \sim \mathbf{sign}(\boldsymbol{W}^{(k)} X_i + \boldsymbol{b} + .1\mathcal{N}(\mathbf{0}, \boldsymbol{I}_M))$, again with $\boldsymbol{b} = -\mathbf{0.5}$. We start with the same coefficients $\boldsymbol{W}^{(0)} = \boldsymbol{I}_M$ and for every changepoint $k > 0$ we rotate the coefficients such that $\boldsymbol{W}_{i,j}^{(k)} = \boldsymbol{W}_{i-1,j}^{(k-1)}$ for $i > 1$ and $\boldsymbol{W}_{1,j}^{(k)} = \boldsymbol{W}_{M,j}^{(k-1)}$. Following Barber et al. (2023), we use two changepoints ($k = 2$) at timesteps 500 and 1500.

3. **Distribution drift:** We follow setting (2) and sample $N = 2000$ i.i.d. data points $(X_i, Y_i)$, with $X_i \overset{\text{iid}}{\sim} \mathcal{N}(\mathbf{0}, \boldsymbol{I}_M)$ and $Y_i \sim \mathbf{sign}(\boldsymbol{W}^{(k)} X_i + \boldsymbol{b} + .1\mathcal{N}(\mathbf{0}, \boldsymbol{I}_M))$, with $\boldsymbol{b}$ as above. Again, we start with $\boldsymbol{W}^{(0)} = \boldsymbol{I}_M$ but now we set $\boldsymbol{W}^{(N)}$ to the last matrix of setting (2). We then compute each intermediate $\boldsymbol{W}^{(k)}$ by linearly interpolating between $\boldsymbol{W}^{(0)}$ and $\boldsymbol{W}^{(N)}$.

After a warmup period of 200 time points, at each time step $n = 200, \ldots, N - 1$ we assign odd indices to the training set, even indices to the calibration set, and we let $X_{n+1}$ be the test point. We fit $M$ independent logistic regression models to the training data to obtain predictors for each label; we let $f_m(X_i)$ denote the estimated probability of the $m^{\text{th}}$ label according to the model. Based on this predictor, we define prediction sets $\mathcal{C}_\lambda(X_i) := \{m \in [M] : f_m(X_i) \geq 1 - \lambda\}$. We compare standard CRC with non-exchangeable (non-X) CRC, for which we use weights $w_i = 0.99^{n+1-i}$ and predict $\hat{\lambda}$ following Eq. (10). In both cases, we minimize the **false negative rate** (FNR):[2]

$$L(\lambda; (X_i, Y_i)) = 1 - \frac{|Y_i \cap \mathcal{C}_\lambda(X_i)|}{|Y_i|}. \tag{14}$$

Note that this loss is nonincreasing in $\lambda$, as required. App. C contains additional experiments considering $\lambda$ to be the number of active labels and using $\mathcal{C}_\lambda(X_i) = \text{top-}\lambda(\boldsymbol{f}(X_i))$.

Fig. 1 shows results averaged across 10 independent trials for $\alpha = 0.2$, summarized in Table 2. We see that the performance of both methods is comparable when the data is i.i.d, with non-X CRC being slightly more conservative. However, when the data is not exchangeable due to the presence of changepoints or distribution drift, our proposed method is considerably better. In particular, after the changepoints in setting (2), non-X CRC is able to achieve the desired risk level more rapidly; in setting (3), the performance of standard CRC gradually drops over time—a problem that can be mitigated by accounting for non-exchangeability introduced by the distribution drift. Importantly, while the average risk is above the predefined threshold for standard CRC for settings (2) and (3) (0.246 and 0.225, respectively), our method achieves the desired risk level on average (0.196 and 0.182, respectively).

## 4.2 MONITORING ELECTRICITY USAGE

We use the ELEC2 dataset (Harries, 1999), which tracks electricity transfer between two states in Australia, considering the subset of the data used by Barber et al. (2023), which contains 3444 time points. The data points correspond to the 09:00am - 12:00pm timeframe and we use the price (`nswprice`, `vicprice`) and demand (`nswdemand`, `vicdemand`) variables as input features,

---

[2]With some abuse of notation, we use $Y_i \subseteq \{1, \ldots, M\}$ to denote the set of gold labels with value $+1$.

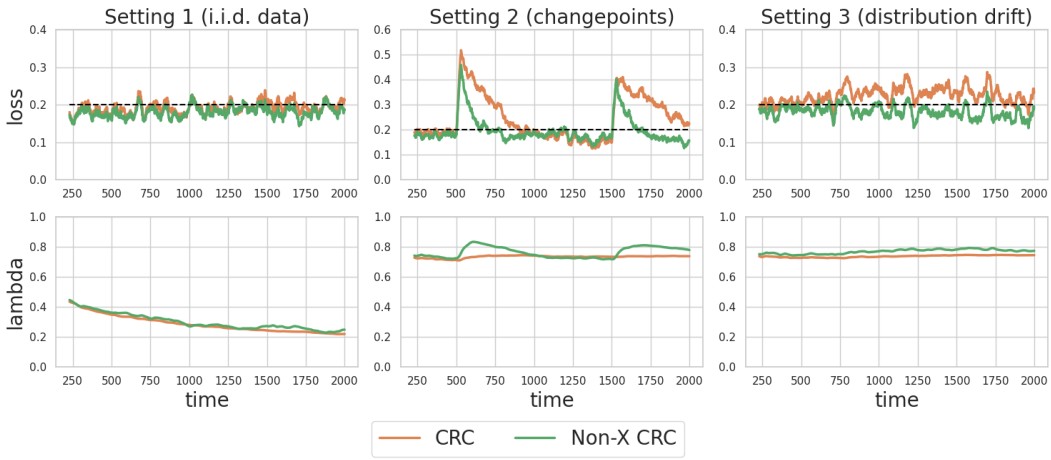

Figure 1: Average loss (top) and $\hat{\lambda}$ (bottom) over 10 independent trials for settings (1), (2), and (3). We smooth all the curves by taking a rolling average with a window of 30 time points.

Table 2: Scalar statistics (mean/median) for settings (1), (2), and (3) for the multilabel classification problem using synthetic time series data reported in §4.1.

| Method | Setting 1 (i.i.d. data) | Setting 2 (changepoints) | Setting 3 (distribution drift) |
|---|---|---|---|
| CRC | 0.191 / 0.183 | 0.246 / 0.228 | 0.225 / 0.218 |
| non-X CRC | **0.181 / 0.175** | **0.196 / 0.183** | **0.182 / 0.175** |

$x_i$ to predict the target `transfer` values $y_i$. We also consider a randomly permuted version of the dataset such that the exchangeability assumption is satisfied. We use the same definitions and settings of §4.1, but this time we fit a least squares regression model to predict the `transfer` values, $\hat{y}_i = f(x_i)$, at each time step. For non-X CRC, we use weights $w_i = 0.99^{n+1-i}$ and we also experiment with weighted least-squares regression, placing weights $t_i = w_i$ on each data point (non-X CRC + WLS). For both standard and non-X CRC we control the residual (distance) with respect to the confidence interval $\mathcal{C}_\lambda(x_i) = [f(x_i) - \lambda, f(x_i) + \lambda]$, where $f(x_i)$ corresponds to the predicted values for `transfer`. We use the $\lambda$-**insensitive absolute loss**, a loss function commonly used in support vector regression (Schölkopf et al., 1998; Vapnik, 1999):

$$L(\lambda; (x_i, y_i)) = \begin{cases} 0, & \text{if } |f(x_i) - y_i| \leq \lambda, \\ |f(x_i) - y_i| - \lambda, & \text{otherwise.} \end{cases} \tag{15}$$

We experiment using $\lambda \in [0, 1]$ with a step of $0.01$. Since we are using the normalized ELEC2 dataset, `transfer` takes values in $[0, 1]$, thus $L(\lambda; (f(x_i), y_i))$ is bounded by $B = 1$. By definition $L(\lambda; (f(x_i), y_i))$ is nonincreasing with respect to $\lambda$.

Fig. 2 shows results for the aforementioned setup. We can observe that in the original setting, both non-exchangeable methods approximate well the desired loss threshold even during the timesteps at which the data suffers from distribution drift. Specifically, as observed by Barber et al. (2023), the electricity *transfer* values are more noisy during the middle of the time range and we can see that the standard CRC + LS method underestimates the $\hat{\lambda}$ for these data points resulting in increased loss, above the desired one. With respect to the CRC + WLS setup, we can see that it manages to reach the desired loss with a smaller interval width on average, indicating that fitting the weighted least-squares model performs better when the data distribution changes, allowing for smaller $\lambda$ during calibration. For the permuted data that simulates the exchangeable data scenario, we can see that all methods perform similarly, reaching the desired loss, as expected.

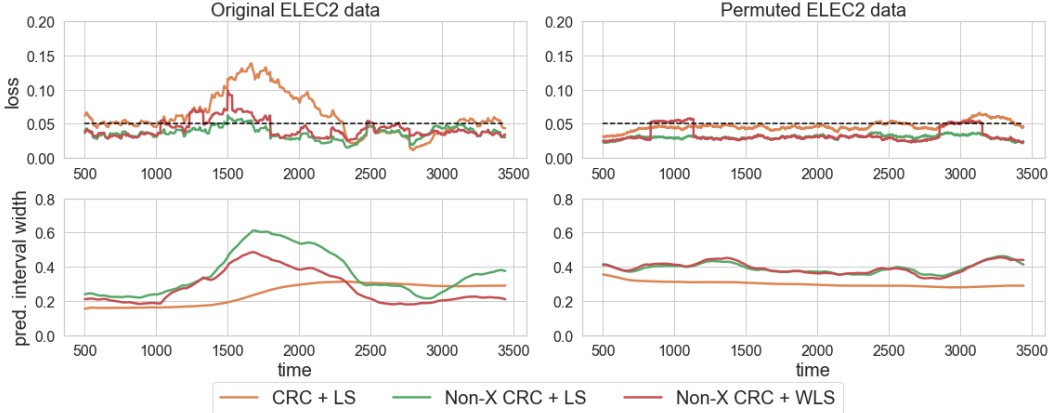

Figure 2: Results on ELEC2 data for $\alpha = 0.05$ and $\lambda$ defined by the prediction interval width. Presented curves are smoothed by taking a rolling average with a window of 300 data points per timestep.

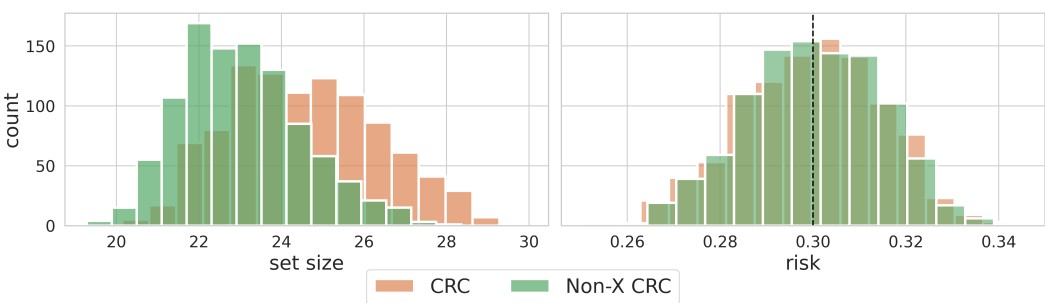

Figure 3: $F_1$-score control on the Natural Questions dataset. Average set size (left) and risk (right) over 1000 independent random data splits.

### 4.3 OPEN-DOMAIN QUESTION ANSWERING

We now shift to open-domain QA, a task that consists in answering factoid questions using a large collection of documents. This is done in two stages, following Angelopoulos et al. (2023a): *(i)* a retriever model (Karpukhin et al., 2020, DPR) selects passages from Wikipedia that might contain the answer to the question, and *(ii)* a reader model examines the retrieved contexts and extract text sub-spans that serve as candidate answers.[3]

Given a vocabulary $\mathcal{V}$, each $X_i \in \mathcal{Z}$ is a question and $Y_i \in \mathcal{Z}^k$ a set of $k$ correct answers, where $\mathcal{Z} := \mathcal{V}^m$ (we assume that $X_i$ and $Y_i$ are sequences composed of up to $m$ tokens). We calibrate the **best token-based** $F_1$**-score** of the prediction set [4], taken over all pairs of predictions and answers,

$$L(\lambda; (X_i, Y_i)) = 1 - \max\{F_1(a, c) : c \in \mathcal{C}_\lambda(X_i), a \in Y_i\}, \qquad \mathcal{C}_\lambda = \{y : f(X_i, y) \geq \lambda\}, \quad (16)$$

which is nonincreasing and upper-bounded by $B = 1$. We consider a CRC baseline that predicts $\hat{\lambda}$ following Eq. (6). For non-X CRC, we choose weights $\{w_i\}_{i=1}^n$ by computing the dot product between the embedding representations of $\{X_i\}_{i=1}^n$ and $X_{n+1}$, obtained using a sentence-transformer model (Reimers & Gurevych, 2019) designed for semantic search,[5] and predict $\hat{\lambda}$ following Eq. (10). While in standard CRC $\hat{\lambda}$ is the same for each test example, this is not the case for non-X CRC.

---

[3]Enumerating all possible answers is intractable, and thus we retrieve the top several hundred candidate answers, extracted from the top 100 passages (which is sufficient to control all risks).

[4]This is the same loss used by Angelopoulos et al. (2023a).

[5]We use the `multi-qa-mpnet-base-dot-v1` model available at `https://huggingface.co/sentence-transformers/multi-qa-mpnet-base-dot-v1`.

While Theorem 1 requires the weights to be independent of the test example, we relax this assumption by setting higher weights for questions in a "neighborhood" of $X_{n+1}$ (see §3). Intuitively, we could think of a situation where the questions are posed by multiple users, each of which may have a tendency to ask semantically similar questions or from the same domain. In this case, we could choose *a priori* higher weights for closer domains/users without violating this assumption.

We use the Natural Questions dataset (Kwiatkowski et al., 2019; Karpukhin et al., 2020), considering $n = 2500$ points for calibration and 1110 for evaluation. Following Angelopoulos et al. (2023a), we use $\alpha = 0.3$ and report results over 1000 trials in Fig. 3. While the test risk is similar in both cases ($0.30 \pm 0.015$), the prediction sets of our method are considerably smaller than those of standard CRC ($23.0 \pm 1.47$ vs. $24.6 \pm 1.83$, respectively). By choosing appropriate weights we can better estimate the set size needed to obtain the desired risk level, while standard CRC tends to overestimate the set size to reach the same value. We thus obtain better estimates of confidence over the predictions.

## 5 RELATED WORK

Conformal prediction (Gammerman et al., 1998; Vovk et al., 1999; Saunders et al., 1999) has proven to be a useful tool for obtaining uncertainty sets/intervals for the predictions of machine learning models, having found a variety of extensions and applications over the years. Among these are *split conformal prediction* (Papadopoulos et al., 2002), which does not require retraining the predictor and instead uses a held-out dataset and *cross-conformal prediction* (Vovk, 2015), which is a hybrid between split conformal prediction and cross-validation. Some of these methods have recently been applied in tasks such as language modeling (Schuster et al., 2022), molecular design (Fannjiang et al., 2022), pose estimation (Yang & Pavone, 2023), and image denoising (Teneggi et al., 2023).

In addition to the works discussed in §2, several extensions to non-exchangeable data have been proposed for time series (Chernozhukov et al., 2018; 2021b; Xu & Xie, 2021; Stankeviciute et al., 2021; Lin et al., 2022; Zaffran et al., 2022; Sun & Yu, 2022; Schlembach et al., 2022; Angelopoulos et al., 2023b), covariate shift (Tibshirani et al., 2019), label shift (Podkopaev & Ramdas, 2021), and others (Cauchois et al., 2020; Gibbs & Candes, 2021; Chernozhukov et al., 2021a; Gibbs & Candès, 2022; Oliveira et al., 2022; Guan, 2022). Moreover, there is recent work aiming at controlling arbitrary risks in an online setting (Feldman et al., 2022). The ideas, assumptions, or formal guarantees in these works are different to ours—we refer the reader to the specific papers for further information.

Angelopoulos et al. (2023a) touch the case of conformal risk control under *covariate shift* (Proposition 3; without providing any empirical validation), explaining how to generalize the work of Tibshirani et al. (2019) to any monotone risk under the strong assumption that the distribution of $Y|X$ is the same for both the training and test data and that the likelihood ratio between $X_{\text{test}}$ and $X_{\text{train}}$ is known or can be accurately estimated using a large set of test data. This result is orthogonal to ours. Besides, they quantify how unweighted conformal risk control degrades when there is an arbitrary distribution shift. Our work is more general and differs in several significant ways: we allow for an arbitrary design of weights, the bounds can be tighter, and the losses are bounded in $[A, B]$, not necessarily in $[0, B]$. Specifically, their Proposition 4 is a particular case of our main result (choosing $A = 0$ and unitary weights), which we use as a baseline in our experiments.

## 6 CONCLUSIONS

We have proposed a new method for conformal risk control, which is still valid when the data is not exchangeable (*e.g.*, due to an arbitrary distribution shift) and provides a tighter bound on the expected loss than that of previous work. Our simulated experiments illustrate how non-exchangeable conformal risk control effectively provides prediction sets satisfying the risk requirements in the presence of non-exchangeable data (in particular, in the presence of change points and distribution drift), without sacrificing performance if the data is in fact exchangeable. Additional experiments with real data validate the usefulness of our approach.

Our work opens up exciting possibilities for research on risk control in challenging settings. For instance, it is an attractive framework for providing guarantees on the predictions of large language models, being of particular interest in tasks involving language generation, medical data (Jalali et al., 2020), or reinforcement learning (Wang et al., 2023), where the i.i.d. assumption does not hold.

## ACKNOWLEDGMENTS

We would like to thank Mário Figueiredo, the SARDINE lab team, and the anonymous reviewers for helpful discussions. This work was built on open-source software; we acknowledge Van Rossum & Drake (2009); Oliphant (2006); Virtanen et al. (2020); Walt et al. (2011); Pedregosa et al. (2011), and Paszke et al. (2019). This work was supported by EU's Horizon Europe Research and Innovation Actions (UTTER, contract 101070631), by the project DECOLLAGE (ERC-2022-CoG 101088763), by the Portuguese Recovery and Resilience Plan through project C645008882-00000055 (Center for Responsible AI), and by Fundação para a Ciência e Tecnologia through contract UIDB/50008/2020.

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

## A   PROOF OF LEMMA 1

The TV distance can be written as an integral probability metric (Müller, 1997):

$$d_{\mathrm{TV}}(P, Q) = \frac{1}{2} \sup_{g:\, \|g\|_\infty \leq 1} \left( \mathbb{E}_P[g] - \mathbb{E}_Q[g] \right). \tag{17}$$

Now, we define $m = (A + B)/2$, $v = (B - A)/2$, and $\bar{f} = (f - m)/v : S \to [-1, 1]$. Noticing that for any $c \in \mathbb{R}$, we have $\mathbb{E}_P[f] - \mathbb{E}_Q[f] = \mathbb{E}_P[f + c] - \mathbb{E}_Q[f + c]$, we can evaluate the difference in expectations as

$$\mathbb{E}_P[f] - \mathbb{E}_Q[f] = v \left(\mathbb{E}_P[\bar{f}] - \mathbb{E}_Q[\bar{f}]\right) \tag{18}$$

$$\leq \frac{B - A}{2} \sup_{g: \|g\|_\infty \leq 1} \left(\mathbb{E}_P[g] - \mathbb{E}_Q[g]\right) \tag{19}$$

$$= (B - A) \, d_{\text{TV}}(P, Q). \tag{20}$$

Repeating with $\bar{f} = (m - f)/v$ (which is also in $[-1, 1]$), yields a similar upper-bound for $\mathbb{E}_Q[f] - \mathbb{E}_P[f]$, from which the result for $|\mathbb{E}_P[f] - \mathbb{E}_Q[f]|$ follows.

## B    PROOF OF THEOREM 1

The proof adapts elements of the proofs from Barber et al. (2023) and Angelopoulos et al. (2023a). Let $Z^K$ be obtained from $Z$ by swapping $(X_K, Y_K)$ and $(X_{n+1}, Y_{n+1})$, where $K$ is a random variable where $\mathbb{P}\{K = i\} = \tilde{w}_i$ (note that $Z^{n+1} = Z$). Let

$$\hat{R}_{n+1}(\lambda) = \sum_{i=1}^{n+1} \tilde{w}_i L(\lambda; (x_i, y_i)) = \frac{N_w \hat{R}_n(\lambda) + L(\lambda; (x_{n+1}, y_{n+1}))}{N_w + 1} \tag{21}$$

be the weighted empirical risk in the calibration set plus the additional test example. Let us define

$$\lambda^* = \inf \left\{\lambda : \hat{R}_{n+1}(\lambda) \leq \alpha\right\}. \tag{22}$$

Given the random variable $Z$, we can think of $\lambda^*(Z)$ as another random variable which is a transformation of $Z$. Moreover, we define the random variable $F_i(Z) = L(\lambda^*(Z); (X_i, Y_i))$ for $i \in [n+1]$, as well as the vector of random variables $F(Z) = [F_1(Z), \dots, F_{n+1}(Z)]$. From Lemma 1, we have

$$\mathbb{E}[F_i(Z^i)] \leq \mathbb{E}[F_i(Z)] + (B - A)d_{\text{TV}}(F(Z), F(Z^i)), \tag{23}$$

a bound that we will use later. Writing $L_i(\lambda) \equiv L(\lambda; (X_i, Y_i))$ for convenience, we also have, for any $\lambda$ and for any $k \in [n+1]$,

$$\hat{R}_{n+1}(\lambda; Z^k) = \sum_{i=1, i \neq k}^{n} \tilde{w}_i L_i(\lambda) + \tilde{w}_k L_{n+1}(\lambda) + \tilde{w}_{n+1} L_k(\lambda)$$

$$= \sum_{i=1, i \neq k}^{n} \tilde{w}_i L_i(\lambda) + \tilde{w}_k (L_k(\lambda) + \underbrace{L_{n+1}(\lambda)}_{\leq B}) + \underbrace{(\tilde{w}_{n+1} - \tilde{w}_k)}_{\geq 0} \underbrace{L_k(\lambda)}_{\leq B}$$

$$\leq \sum_{i=1, i \neq k}^{n} \tilde{w}_i L_i(\lambda) + \tilde{w}_k (L_k(\lambda) + B) + (\tilde{w}_{n+1} - \tilde{w}_k)B$$

$$= \sum_{i=1}^{n} \tilde{w}_i L_i(\lambda) + \tilde{w}_{n+1} B$$

$$= \frac{N_w}{N_w + 1} \hat{R}_n(\lambda; Z) + \frac{B}{N_w + 1}. \tag{24}$$

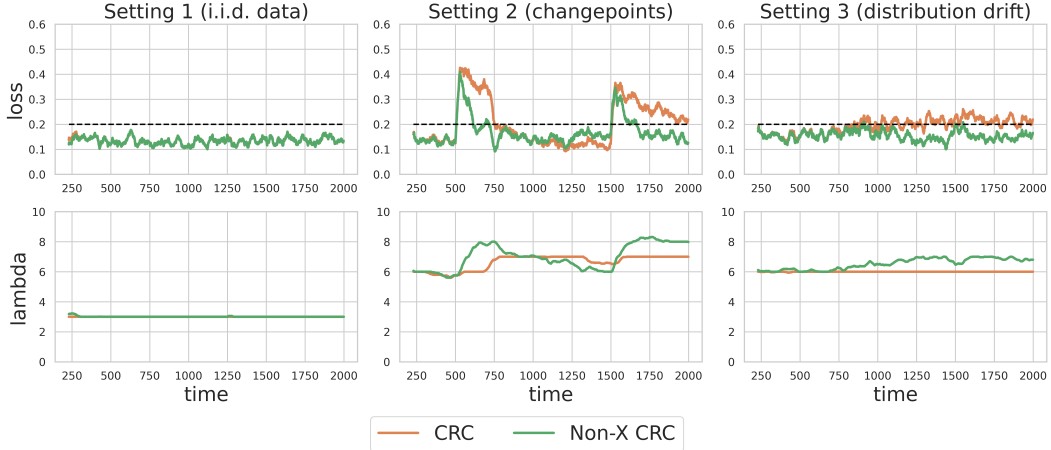

Figure 4: Average loss (top) and $\hat{\lambda}$ (bottom) over 10 independent trials for settings (1), (2), and (3). In this case, $\lambda$ represents the number of predicted labels. We smooth the curves by taking a rolling average with a window of 30 time points.

Therefore, setting $\lambda = \hat{\lambda}$ and using Eq. (10), we obtain $\hat{R}_{n+1}(\hat{\lambda}; Z^k) \leq \frac{N_w}{N_w+1}\hat{R}_n(\hat{\lambda}; Z) + \frac{B}{N_w+1} \leq \alpha$, which, from Eq. (22), implies $\lambda^*(Z^k) \leq \hat{\lambda}(Z)$. Since the loss $L$ is nonincreasing with $\lambda$, we get

$$
\begin{aligned}
\mathbb{E}[L_{n+1}(\hat{\lambda}(Z); Z)] &\leq \mathbb{E}[L_{n+1}(\lambda^*(Z^K); Z)] = \mathbb{E}[L_K(\lambda^*(Z^K); Z^K] \\
&= \sum_{i=1}^{n+1} \underbrace{\mathbb{P}\{K=i\}}_{=\tilde{w}_i} \underbrace{\mathbb{E}[L_i(\lambda^*(Z^i), Z^i]}_{=\mathbb{E}[F_i(Z^i)]} \\
&\leq \sum_{i=1}^{n+1} \tilde{w}_i \left( \underbrace{\mathbb{E}[L_i(\lambda^*(Z), Z]}_{=\mathbb{E}[F_i(Z)]} + (B-A)d_{\text{TV}}(F(Z), F(Z^i)) \right) \\
&= \mathbb{E}\left[ \sum_{i=1}^{n+1} \tilde{w}_i L_i(\lambda^*(Z), Z) \right] + (B-A)\sum_{i=1}^{n} \tilde{w}_i d_{\text{TV}}(F(Z), F(Z^i)) \\
&= \mathbb{E}\left[ \hat{R}_{n+1}(\lambda^*(Z)) \right] + (B-A)\sum_{i=1}^{n} \tilde{w}_i d_{\text{TV}}(F(Z), F(Z^i)) \\
&\leq \alpha + (B-A)\sum_{i=1}^{n} \tilde{w}_i d_{\text{TV}}(F(Z), F(Z^i)). \quad (25)
\end{aligned}
$$

The result follows by noting that $d_{\text{TV}}(F(Z), F(Z^i)) \leq d_{\text{TV}}(Z, Z^i)$. Eq. (25) is actually a tighter bound, similarly to what has been noted by Barber et al., 2023.

## C    MULTILABEL CLASSIFICATION IN A TIME SERIES

Fig. 4 shows results averaged across 10 independent trials for $\alpha = 0.2$ and setting $\lambda$ in a slightly different way than that of §4.1. In this case, $\lambda$ represents the number of active labels and we use $\mathcal{C}_\lambda(X_i) = \text{top-}\lambda(\boldsymbol{f}(X_i))$. The main takeaways remain the same: both methods perform similarly when the data is exchangeable, in setting (1). Accounting for the non-exchangeability introduced by changepoints and distribution drift using our method enables lowering the risk to the desired level in settings (2) and (3).

