# OpenReview forum: "Non-Exchangeable Conformal Risk Control"
_ICLR.cc/2024/Conference — ICLR 2024 poster_

### Official Review · Reviewer_r1o1 · 2023-10-30

**Soundness:** 3 good
**Presentation:** 3 good
**Contribution:** 2 fair
**Rating:** 6
**Confidence:** 4

**Summary:**

The paper shows how to perform conformal risk in a non-exchangeable setting --- i.e. for any given loss l, providing a threshold such that the true loss is less than some pre-specified alpha. Unlike the typical conformal prediction setting that assumes that data satisfies the exchangeable assumption (i. e. any permutation of the data is equally likely), it tries to relax such assumption. Also, unlike the typical conformal prediction that provides a prediction set that's supposed to contain the true label, it studies a more general form usually referred to as conformal risk control.

The paper achieves the above goal by combining Barber et al. (Conformal Prediction Beyond Exchangeability) and Angelpoulos et al. (Conformal Risk Control).

Finally, the paper tests the idea on three datasets.

**Strengths:**

-The idea proposed here seems to actually work in practice as shown by their experiments.

**Weaknesses:**

-The main contribution of the paper seems to be just a combination of two techniques, conformal prediction beyond exchangeability and conformal risk control. Once one understand the idea behind conformal prediction (i.e. finding an (1-alpha) quantile of the conformal score in the calibration data set), the idea of conformal risk control follows pretty naturally (finding some alpha-cut off for some monotonic loss in the calibration data set). And hence, techniques known for conformal prediction can be easily translated to conformal risk control — i.e. handling distribution shift. In this specific case, the paper is leveraging the technique of conformal prediction beyond exchangeability. Also, there isn’t inherent difficulty in applying the beyond the exchangeability idea to this conformal risk control setting to my knowledge. If there is additional difficulty in applying the beyond the exchangeability idea to the conformal risk control setting compared to applying the idea to the typical conformal prediction setting, it would be great if the authors can emphasize that and how this difficulty was overcome.
-There’s no clear guideline as to how to go about setting these weights and one has to resort to heuristics. But I think this point is not necessarily specific to the approach in this paper but with the original conformal prediction beyond exchangeability of Barber et al.

**Questions:**

-In Equation (3) where Z_i is defined, shouldn’t Z_i be what you get by swapping the test point (X_{n+1}, y_{n+1}) with (X_i, y_i) as opposed to the nth calibration data point (X_n, y_n)? This is how things are defined in Barber et al. as well.

---

> ### Author Response · Authors · 2023-11-17
>
> Thank you for your review. We address below your concerns about our paper.
>
> > The main contribution of the paper seems to be just a combination of two techniques, conformal prediction beyond exchangeability and conformal risk control (...) techniques known for conformal prediction can be easily translated to conformal risk control — i.e. handling distribution shift. (...) there isn’t inherent difficulty in applying the beyond the exchangeability idea to this conformal risk control setting to my knowledge. If there is additional difficulty (...) it would be great if the authors can emphasize that and how this difficulty was overcome.
>
> We respectfully disagree with the claim that our paper is “just” a combination of two techniques and that “there isn’t inherent difficulty in applying the beyond the exchangeability idea to this conformal risk control setting”. For example, one nontrivial aspect in our proof was to generalize / replace the step in Barber et al.'s proof (their Eqs. 24-26) which involves defining the set of “strange” points and showing that noncoverage implies strangeness of point K. In conformal risk control, “noncoverage of a test point” does not make sense, since we are upper bounding the risk and not modeling coverage. To overcome this difficulty and prove our theorem in the context of risk control, we handled this by (1) defining random variables $F(Z)$ which contain the losses as a function of $\lambda^*(Z)$, and (2) deriving a bound that includes a TV distance using $F(Z)$, and (3) invoking the non-monotonicity of the loss to obtain the final result. These elements are specific to our paper and absent in Barber et al. (2023) and Angelopoulos et al. (2023)’s proofs. In short, the proof style is similar but some of the key steps are substantially different and nontrivial. We invite the reviewer to compare the proofs and check the differences in case any doubt remains.
>
> As pointed out by Reviewer wGbj, non-exchangeability happens frequently in practice, and thus it is important to explore robust weighting schemes and their implications. As explained, our method and proofs use techniques from Barber et al. (2023) and Angelopoulos et al. (2023), but their combination is not straightforward and has not been done before. We address the problem of risk control under non-exchangeable settings, which is highly relevant, yet the existing literature doesn't offer comprehensive solutions (please see Reviewer 2JpW’s comment on this topic). In addition to the theoretical results, our experiments show that the proposed method is useful in practice in both synthetic and real-world settings.
>
> We hope this clarifies our goals and the significance of our contributions and alleviates your concerns.
>
> > There’s no clear guideline as to how to go about setting these weights and one has to resort to heuristics. But I think this point is not necessarily specific to the approach in this paper but with the original conformal prediction beyond exchangeability of Barber et al.
>
> As pointed out in the answer to Reviewer wGbj, although our theoretical bound on $\mathbb{E}[L(\hat{\lambda}; (X_{n+1}, Y_{n+1}))]$ holds for any choice of weights, its usefulness relies on the coverage gap in Eq. 7 being small, i.e., on our ability  to choose small weights $w_i$ for data points $Z^i$ with large total variation distance $d_{\mathrm{TV}}(Z, Z^i)$. In Barber et al. 2023 (Lemma 1) it is shown that $d_\mathrm{TV}(Z, Z^i) \le 2d_\mathrm{TV}(Z_i, Z_{n+1})$, and while the true value of this term is unknown, in some cases it can be properly modeled or approximated. For instance, in the presence of distribution drift in time series, we expect  $d_{\mathrm{TV}}(Z_i, Z_{n+1})$ to decrease with $i$, therefore we should  choose weights that increase with $i$. Our experiments (e.g., $\S4.1$, Setting 3) empirically validate this assumption for the case of distribution drift. The same principle can be applied in other domains (e.g., for spatial data, one may place higher weights to points close in space to the test point).
>
> We agree  that the paper benefits from more discussion on this topic, and for that reason we updated it accordingly (please see $\S3$ after Eq. 8). In particular, we included a new subsection ($\S3.2$) that suggests a practical strategy for choosing the weights based on knowledge or assumptions about how the TV distance $d_{\mathrm{TV}}(Z, Z^i)$ varies over time, via a regularized minimization of the coverage gap using the maximum entropy principle. We also show in this subsection that this principle provides further justification for the use of weights with an exponential decay in time series under a distribution shift, which we use in our experiments of $\S4.1$ and $\S4.2$. We hope this addresses your main concerns.
>
> _(continues in a follow-up comment)_

---

> > ### Author Response · Authors · 2023-11-17
> >
> > > In Equation (3) where Z_i is defined, shouldn’t Z_i be what you get by swapping the test point (X_{n+1}, y_{n+1}) with (X_i, y_i) as opposed to the nth calibration data point (X_n, y_n)? This is how things are defined in Barber et al. as well.
> >
> > While our Theorem 1 had the correct definition (below Eq. 12), there was in fact a typo in Eq. 3, which we already fixed. Thanks for pointing this out.
> >
> > Please let us know if our answers alleviate your concerns.

---

> ### Comment · Reviewer_r1o1 · 2023-11-18
> **Thanks for the response!**
>
> (1) Novelty beyond Barber et al. and Angelopoulos et. al.
>
> Maybe this has to do with me not having read Barber et al. and Angelopoulos et al. very closely, and I’m missing some nuances in technical details. I understand the proof that is described in the response is not in Barber et al. and additional arguments will be needed to handle conformal risk control aspect of the paper. However, my understanding was that what is in the described proof sketch in the response is essentially the main insights in Angelopoulos et al.
>
> Let me try to clarify with more specific details, and please correct me if I’m wrong.
>
> As said in the response, talking about non-coverage of a test point doesn’t make sense the conformal risk control, and the set of “strange points” has to be thought through more carefully. And the original conformal risk control paper shows how one can define a set of strange points not just via non-coverage but via thresholding a monotonic loss, thereby going from vanilla conformal prediction to conformal risk control. The insight here I guess is realizing that given a monotonic non-conformal score, the prediction set is monotonic in the threshold.
>
> Therefore, one can just go to the proof that’s outlined on page 32 of Barber et al. (right before section 7) and instead of miscoverage (a point in the prediction set or not), one can write down the monotonic loss and the rest of the proof follows; my impression is that this swapping of miscoverage with a monotonic loss has been already done in Angelopoulos et al.
>
> In fact, the proof of Theorem 1 of this paper looks like the proof of Theorem 2 in Barber et al. (specifically the proof on page 32 right before section 7) except for the swapping of the miscoverage with the monotonic loss. And one can see this connection of miscoverage and monotonic loss in Section 2 of Angelopoulos et al.
>
> Let me know if I'm missing more subtle nuances.
>
>
>
> (2) Choice of Weights
>
> Thank you for adding more discussion on this topic of how to go about choosing these weights.

---

> > ### Author Response · Authors · 2023-11-18
> > **Detailed comparison of our proof with that of Barber et al.**
> >
> > We appreciate your quick response and the clarification of your question. However, we insist that our proof is **not** just a trivial extension of Barber el al. obtained by replacing miscoverage by a monotonic loss. Of course, we use elements of both papers (Barber et al. and Angelopoulos et al.) which were a great inspiration for you work, and as we said in our previous response the proof style is similar to Barber et al., but again our proof has important and subtle differences which we detail below step by step, in the hope that this helps clarifying.
> > - Barber et al. (section 6.2) provides a proof for full CP of which split CP is a special case. We drop here some of their extra notation which is specific to full CP and translate their proof to the split CP case, which is what concerns us. In split CP the function $\hat{\mu}^k$ is fixed (fit on a separate training set), so we can write the residuals simply as $R_1, ..., R_{n+1}$ and drop the superscripts. They define (in their eq. 20) the random variables $R(Z^k)$ where $Z^k$ is the sequence where the data points $k$ and $n+1$ are swapped. They relate miscoverage with the event that $R_{n+1} > Q_{1-\alpha}(\sum_{i=1}^{n}w_i \delta_{R_i} + w_{n+1}\delta_{+\infty})$. Note that there are two ``loss functions'' here: the one associated to miscoverage (a 0/1 loss) and the one associated to the residuals (in their case the absolute error $R_i = |y_{i} - \hat{\mu}^k(x_i)|$).
> >   - **In our case there is one loss playing both roles ($L(\lambda, Z_i)$). As you say, in the exchangeable case Angelopoulos  generalizes CP to CRC by introducing this loss (CP corresponding to $L$ being 0/1). However, in the nonexchangeable case (our paper) the TV term in the coverage gap includes also this loss.** This loss depends on $\lambda$ (unlike the residuals in Barber et al., which do not have a similar dependency). To handle this, we have to consider the random variables $Z$, the random variable $\lambda^*(Z)$ (which we can think of as the "optimal" $\lambda$ in hindsight), and the random variables $F(Z) = [L_1(\lambda^*(Z)), ..., L_{n+1}(\lambda^*(Z))]$. These random variables are all transformations of each other. This construction is different from the one in Barber et al. and it is absent in Angelopoulos et al.
> > - Barber et al. proceed to show the bound (their eq. 22) $Q_{1-\alpha}(\sum_{i=1}^n \delta_{R_i} + w_{n+1}\delta_{+\infty}) \ge Q_{1-\alpha}(\sum_{i=1}^{n+1} \delta_{R_i(Z^k)})$.
> >   - This corresponds to our bound (our eq. 24) $R_{n+1}(\lambda, Z^K) \le \frac{N_w}{N_{w}+1} R_{n+1}(\lambda, Z) + \frac{B}{N_w+1}$. (Note that despite using the same letter our $R_i$ is the weighted risk, different from the residual $R_i$ in Barber et al.)
> > - They use their bound to show that miscoverage implies $R_k(Z^k) > Q_{1-\alpha}(\sum_{i=1}^{n+1} \delta_{R_i(Z^k)})$ (their eq. 23).
> >   - We use our bound to show that $R_{n+1}(\hat{\lambda}, Z^k) \le \alpha$ implies $\lambda^*(Z^k) \le \hat{\lambda}(Z)$. This is qualitatively different.
> > - They define the set of strange points $\mathcal{S}(r)$ as a function of residuals $r=(r_1, ..., r_{n+1})$ and show that $\sum_{i \in \mathcal{S}(r)} w_i \le \alpha$.
> >   - We don't need this in our proof.
> > - From their eq. 23 they show that miscoverage implies $k \in \mathcal{S}(R(Z^k))$.
> >   - The corresponding result in our proof is $\mathbb{E}[L_{n+1}(\hat{\lambda}(Z); Z)] \le \mathbb{E}[L_{k}(\lambda^*(Z^K); Z^K)]$ (first line of our eq. 25). Philosophically similar but different -- we have the same loss in both sides, which depends on $\lambda$, and two different $\lambda$ in this expression.
> > - Then their eq. 27 bounds $P(K \in \mathcal{S}(R(Z^K)))$ in a sequence of steps.
> >   - We work with the expected loss instead. But our steps are similar to theirs here. The main difference is that we need to invoke our Lemma 1 in one of the steps to obtain the desired TV term.
> >
> > We sincerely hope that the explanation above may convince you that our theoretical result and its proof are a bit more than just "stapling" together the results of Barber et al. and Angelopoulos et al. (even though, as we acknowledge, these two works have been very inspirational to us). We are of course happy to clarify any other concerns you might have.
> >
> > In any case, and independently of the proof being easier or harder, we believe that the most important is its technical correctness. In the absence of technical flaws, what we are stating is a new result that links and complements the two lines of work above. We believe this is an important (and novel) result per se, and as you point out in your review, "seems to actually work in practice as shown by their [ours] experiments."

---

> > > ### Comment · Reviewer_r1o1 · 2023-11-21
> > > **Thanks for the detailed comparison**
> > >
> > > First of all, I really wish I took out the word "just" when describing the combination of ideas as i see that it might have come across as dismissive of the work. So my apologies. But I do think that when evaluating a paper, it isn’t too inappropriate to consider the novelty of the paper on top of technical correctness.
> > >
> > > In the above response, I have some questions regarding the response:
> > > 1. “unlike the residuals in Barber et al., which do not have a similar dependency". Doesn't Barber et al., choose lambda to be essentially the alpha quantile of the residuals or equivalently conformal scores of the calibration dataset too?
> > >
> > > More specifically, I thought that since we are talking about split conformal prediction where the predictor $\mu$ is fixed, there's no difference in analysis in terms of studying the distribution over the original points $(x,y)$ and and studying the distribution over the residuals $(x, |\mu(x) - y|)$ or any other conformal score $(x, s_\mu(x,y))$, no? More specifically, given a calibration dataset $((x_1, y_i), ..., (x_n, y_n))$, there's an immediate deterministic mapping to $((x_1, s_\mu(x_1,y_1)), ... , (x_n, s_\mu(x_n, y_n)))$ if you fix the conformal score function $s$ and the predictor $\mu$ beforehand.
> > >
> > > Therefore, even the TV distance between the distribution over (x,y)’s should be the same for the distribution over (x,s)’s, right?
> > >
> > > So I don’t understand this comment about two loss functions? To me, there’s only one loss function in Barber et al. too in the split conformal case.: in fact, to me the insight of Angelopoulos et al is  you get conformal prediction if you use the loss function as $E_{(x,y)}[1[s(x,y) \le \lambda]]$ which is monotonic in lambda.
> > >
> > >
> > > 2. “They show their bound to show that miscoverage implies …(their eq 23) … We use our bound to show that ... . This is qualitatively different".
> > > First, just for me to understand what is stated in the response: it’s saying that if the expected risk of the (n+1)th point was less than alpha, then it must have been the case that the chosen $\hat{\lambda}$ must have been big enough compared to the true cut-off $\lambda^*$ to guarantee that the expected risk of
> > >
> > > Can you state the qualitative difference here to Barber et al. a little more? The bound that this paper talks about in terms of comparing $\hat{\lambda}$ and $\lambda^*$ is rather similar to the inequality 22 in Barber et al.? The left-hand side of the inequality is like $\hat{\lambda}$ and the right hand side inequality is like $\lambda*$.
> > >
> > >
> > > Anyway, I have changed my score in light of thinking through this a little more.

---

> > > > ### Author Response · Authors · 2023-11-22
> > > > **Thank you for updating the score**
> > > >
> > > > Thank you for your answer and for updating your score!
> > > >
> > > > Regarding your questions:
> > > >
> > > > > “unlike the residuals in Barber et al., which do not have a similar dependency". Doesn't Barber et al., choose lambda to be essentially the alpha quantile of the residuals or equivalently conformal scores of the calibration dataset too?
> > > >
> > > > Sorry if we weren't clear in our response. What we meant is that in Barber et al. their coverage gap is $d_{TV}(R(Z), R(Z^i))$, where $R$ are the residuals, which depend on the points $(x,y)$ and the predictor function $\mu$. Our coverage gap, on the other hand, is $d_{TV}(F(Z), F(Z^i))$, which depends on analogous quantities but also on $\lambda^*$. While $\lambda^*$ is of course a function of the data, this is qualitatively different from Barber et al. -- their $R(Z)$ does not involve any calibrated quantity (and also not the alpha quantile).
> > > >
> > > > > "even the TV distance between the distribution over (x,y)’s should be the same for the distribution over (x,s)’s, right?"
> > > >
> > > > No, in fact we have $d_{TV}(Z, Z^i) \ge d_{TV}(s(Z), s(Z^i))$ for any function s. This is because TV distance is a f-divergence and therefore satisfies the data processing inequality: transforming random variables through any function can only *decrease* the TV distance. See e.g. ref [1], Fact 1.1, p 4 in the preprint https://ccanonne.github.io/survey-topics-dt.html. In fact, Barber et al. also have this bound (when s is the residuals) in their p. 15 after Theorem 2.
> > > >
> > > > [1] Clément L. Canonne (2022), "Topics and Techniques in Distribution Testing: A Biased but Representative Sample", Foundations and Trends® in Communications and Information Theory: Vol. 19: No. 6, pp 1032-1198
> > > >
> > > > > Can you state the qualitative difference here to Barber et al. a little more? The bound that this paper talks about in terms of comparing $\hat{\lambda}$ and $\lambda^*$ is rather similar to the inequality 22 in Barber et al.? The left-hand side of the inequality is like $\hat{\lambda}$ and the right hand side inequality is like $\lambda^*$.
> > > >
> > > > Close, but not quite. The left-hand side of the inequality 22 in Barber et al. corresponds to $\frac{N_w}{N_w+1} R_{n+1}(\lambda, Z) + \frac{B}{N_w+1}$ and the right hand side corresponds to $R_{n+1}(\lambda, Z^K)$. We need to use the definition in our eq. 22 and the monotonicity of the losses to conclude that $\lambda^*(Z^K) \le \hat{\lambda}(Z)$.

---

> > > > > ### Comment · Reviewer_r1o1 · 2023-11-22
> > > > > **Thanks for the clarification**
> > > > >
> > > > > Thanks for the clarification. I think I finally understand the the paper's subtle difference to Barber et al. and Angelopoulos et. al.: the TV-distance discussion above (in terms of its presence/absence of the lambdas) helped me.
> > > > >
> > > > > I have raised the score accordingly.

---

> > > > > > ### Author Response · Authors · 2023-11-23
> > > > > > **Thank you**
> > > > > >
> > > > > > Thank you for your time and effort reviewing and discussing our paper! We are glad that we clarified the difference and addressed your main concern.

---

### Official Review · Reviewer_aj7z · 2023-10-31

**Soundness:** 2 fair
**Presentation:** 2 fair
**Contribution:** 2 fair
**Rating:** 5
**Confidence:** 4

**Summary:**

This paper extends conformal risk control (CRC) under the exchangeable setup to the non-exchangeable setup, i.e., converting “Angelopoulos et al. (2023a)” to the “Barber et al. (2023)”-style. In particular, the coverage guarantee under non-exchangeability is stated and proven in Theorem 1 and the proof follows techniques by Barber et al. (2023) and Angelopoulos et al. (2023a). The efficacy of the proposed algorithm is empirically demonstrated by using synthetic and real data with multiple shifts.

**Strengths:**

Originality: This paper combines the results by Barber et al. (2023) and Angelopoulos et al. (2023a), leading to a new result.

Quality: the claim is well-justified via Theorem 1 and its proof.

Clairity: the paper is mostly well-written.

Significance: considering that the conformal prediction can be extended to the non-exchangeable setup by Barber et al. (2023), so it is not surprising that conformal risk control can be extended in a similar way. But, it is still a new result.

**Weaknesses:**

The following includes my concerns.

1. Under the non-exchangeable setup, the CRC should be broken, and this is why we need non-exchangeable extension of CRC. But, I cannot see the trend in Setting 3 in Figure 1 and Figure 3, which is unsatisfactory. In particular, I’m not convinced why open-domain QA experiments (related to Figure 3) fit the non-exchangeable setup – the concrete scenario on why we need to consider the non-exchangeable setup here is required. Moreover, the way to generate w_i is not correct – by Barber et al. (2023), w_is prespecified but not data-dependent (see Section 4.5. by Barber et al. (2023) for a careful discussion on the data-dependent weights). This may demonstrate that open-domain QA is not a proper target of this method.

2. It would be more readable if the controlled loss is summarized in scalar statistics. For example, in Setting 3 in Figure 1, I cannot see why the proposed approach is good.

**Questions:**

1. Can you draw plots such that CRC is clearly broken under the non-exchangeable setup?
2. Can you provide the concrete scenario on why we need to consider the non-exchangeable setup in open-domain QA experiments? You also can replace the experiments.
3. Can you justify why weights are generated in a data-dependent way in open-domain QA experiments?
4. Can you summarize / provide scalar statistics on the controlled loss such that we can see whether the proposed approach controls the risk but the baseline fails to control it?

---

> ### Author Response · Authors · 2023-11-17
>
> Thank you for your thoughtful review and suggestions. We address the main points below.
>
> > Under the non-exchangeable setup, the CRC should be broken (...); (...) in Setting 3 in Figure 1, I cannot see why the proposed approach is good.; Can you summarize / provide scalar statistics on the controlled loss such that we can see whether the proposed approach controls the risk but the baseline fails to control it?
>
> Please note that Figure 1 shows that the CRC baseline fails at providing the desired risk level in the presence of distribution drift (setting 3), whereas the proposed approach does not: the orange curve (CRC) is mostly above the black dotted line, while the green one (our method) tends to be below. We rescaled the axes in  this figure for clarity. We also followed your suggestion and  added scalar statistics  (mean/median) to the paper, which further strengthen this point  (see Table 2, reproduced  below). We also included some of these values in the main text (last paragraph of $\S4.1$) for further clarity. We hope to have clarified  your concern.
>
> | Setting   | 1 (i.i.d data) | 2 (changepoints) | 3 (distribution drift) |
> |-----------|----------------|------------------|------------------------|
> | CRC       | 0.191/0.183    | 0.246/0.228      | 0.225/0.218            |
> | Non-X CRC | 0.181/0.175    | 0.196/0.183      | 0.182/0.175            |
>
> > In particular, I’m not convinced why open-domain QA experiments (related to Figure 3) fit the non-exchangeable setup (...) the way to generate w_i is not correct – by Barber et al. (2023), w_is prespecified but not data-dependent (...); Can you justify why weights are generated in a data-dependent way in open-domain QA experiments?
>
> For an intuitive scenario where non-exchangeability provides a good framework for the QA experiment, imagine a situation where the questions are posed by multiple users, with the same user having a tendency to ask semantically similar questions. Our results in this experiment suggest that choosing weights based on question similarity leads to smaller prediction sets while attaining the same risk level (see Figure 3 (left)). We believe this is an interesting result potentially useful in future research.
>
> You are right that Theorem 1 requires the independence $K \perp Z$ as in Barber et al. (2023). For the other experiments ($\S4.1$ and $\S4.2$), we always assume data-independent weights. Our goal with the QA experiment is to assess whether the framework still works in practice when this assumption is relaxed (in cases where prior information about the domain of the questions or the user who asked them is available, one could choose a priori higher weights for closer domains/users without violating this assumption). Furthermore, note that it is possible to generalize our bound to allow data-dependent weights by conditioning the TV distance on the weights as described by Barber et al. (2023), $\S4.5$ (paragraph “Fixed versus data-dependent weights'').
>
> To sum up, we believe that the results of $\S4.3$ are still interesting and useful. We agree, however, that the paper would benefit from a clearer discussion about this point along with a justification for that experiment. Therefore, we  added this information in $\S3$ (last paragraph before $\S3.1$) and $\S4.3$ and updated the abstract/introduction accordingly. We hope this alleviates your concern.

---

> > ### Author Response · Authors · 2023-11-21
> >
> > Thank you for your time reviewing our paper. We hope you had the chance to look at our answers and updated version of the paper. Since the author response period is almost ending, we kindly request your feedback on wether we addressed your concerns. Please do let us know if further clarification is needed.

---

> > ### Comment · Reviewer_aj7z · 2023-11-23
> > **Thanks**
> >
> > Thanks for detailed response! The response enhanced my understanding, but I still have concerns.
> >
> > 1. To my understanding, the authors' may be confused the empirical realization of the marginal coverage guarantee (5). As it is not conditioned on the calibration set (i.e., the PAC-style guarantee), the empirical coverage should be *around* alpha (not below alpha) -- if it is not convinced, see Figure 2 of Tibshirani et al. (2019). Thus, Table 2 does not demonstrate non-X CRC is clearly better than CRC (as the empirical risks are around alpha) under shifts.
> >
> > 2. It is okay to show practical values of the proposed method in open-domain QA experiments, but anyway it is heuristic and not supported by the main theorem, which even undermine the value of the main theorem. I'd replace this into another experiment that justifies the main theorem, and put open-domain QA experiments into the other section to independently highlight the claimed practical value.
> >
> > Mainly the empirical results do not support the paper's claim, so, I'll maintain my score.

---

> > > ### Author Response · Authors · 2023-11-23
> > >
> > > Thanks for reading our response. We answer both of your points below:
> > >
> > > > the empirical coverage should be around alpha (not below alpha) (...) if it is not convinced, see Figure 2 of Tibshirani et al. (2019)
> > >
> > > We respectfully disagree. Our Theorem 1 should be interpreted as providing an **upper bound** to the expected test loss, not an estimate of this test loss. In many practical problems where risk is critical, it is important to have such upper bounds. By “around alpha” perhaps you mean that the extra term in our upper bound allows some slack and therefore the expected loss could potentially be a bit above alpha? However, you might be missing that  the main point of our experiment is that if the weights are chosen wisely (see the added section 3.2), then this extra term will be small and we should expect non-X CRC to have a smaller upper bound than CRC (which corresponds to the naive choice of uniform weights). This matches what we observe in Table 2. Note in particular that for setting 2 we have  for 0.246 for CRC vs 0.196 for non-X CRC, with $\alpha=0.2$. We think this clearly shows our point.
> > >
> > > We do not understand your point about Figure 2 of Tibshirani et al. (2019) (https://arxiv.org/pdf/1904.06019.pdf). In fact, this figure seems to further support our point. Note that Tibshirani et al. is working on a CP setting where they seek a coverage of **at least** 90%, whereas we are working on a  CRC setting where we seek an expected loss of **at most** 0.2 (that is, $\alpha$ is a **lower bound** for them but an **upper bound** for us). You can see that in their Figure 2 all the weighted conformal predictors give above 90% coverage, just like in our case our weighted CRC gives expected losses below 0.2. Likewise, their vanilla CP baseline (top figure, in blue) fails by attaining only 0.82 coverage, the same way as our vanilla CRC baseline fails by exceeding the expected loss 0.2 (considerably for setting 2, and by a 10% factor in setting 3).
> > >
> > > > It is okay to show practical values of the proposed method in open-domain QA experiments, but anyway it is heuristic and not supported by the main theorem, which even undermine the value of the main theorem. I'd replace this into another experiment that justifies the main theorem, and put open-domain QA experiments into the other section to independently highlight the claimed practical value.
> > >
> > > Please note that we **already** have other experiments (in $\S4.1$, which uses synthetic data and in $\S4.2$, which uses real data) where weights are data-independent. These experiments already justify the main theorem, as argued above and in our previous answer. Therefore, we do not think more experiments are necessary to illustrate the theorem. However, we agree with your suggestion of moving the open-domain QA experiments to a separate section on data-dependent weights, turning section 4.3 into section 5 and renaming it. We don't have time to update our manuscript before the rebuttal deadline (in less than 1h) but we are happy to do this for the camera ready.
> > >
> > > I hope this alleviates all your remaining concerns.

---

> > > > ### Comment · Reviewer_aj7z · 2023-12-02
> > > >
> > > > Thanks for the answers.
> > > >
> > > > On the evaluation results in Table 2:
> > > > First of all, I assume that this paper follows a conventional way for evaluation (as I couldn't find details): (1) split a calibration set and test set, (2) use the calibration set for constructing the conformal predictor, and (3) use the constructed conformal predictor to compute the empirical coverage for all test samples --- note that if you draw independent calibration sets for *each* test sample for your Table 2, I'd be happy with the result (even though it is a quite impractical setup).
> > > >
> > > > Under my assumption, I think the authors' do not correctly distinguish between the theoretical meaning of the coverage guarantee and the empirical coverage guarantee under the above conventional way for evaluation. First, the theoretical coverage guarantee is not the guarantee conditioned on a calibration set (for the calibration-set conditional guarantee, i.e., the PAC guarantee, see Section 3 of [R1]). Even though, the theoretical guarantee is not conditioned on the calibration set, but in the experiments, we only have a hold-out calibration set. So, we unfortunately need to compute the empirical coverage over all test samples given a single calibration set. That's why the empirical coverage is around the desired coverage, as shown in Figure 2 of Tibshirani et al. (2019).
> > > >
> > > > Considering this, the empirical coverages in Table 2 for both CRC and non-X CRC are around the desired coverage, meaning that we cannot confirm that the proposed method is effective under distribution shifts. This was my main concern on the efficacy of the proposed method (and may require different experiments to contrast your method).
> > > >
> > > >
> > > > [R1] https://proceedings.mlr.press/v25/vovk12/vovk12.pdf

---

### Official Review · Reviewer_wGbj · 2023-11-01

**Soundness:** 3 good
**Presentation:** 3 good
**Contribution:** 2 fair
**Rating:** 5
**Confidence:** 5

**Summary:**

This paper expands on the non-exchangeable setting for conformal risk control by Angelopoulos et. al. (2023). Conformal risk control is a generalization of conformal prediction that expands control of coverage losses to control of any monotonic, upper-bounded, exchangeable function $L \colon \Lambda \rightarrow [0, B]$, where $\Lambda$ is the space of single-dimensional inputs to ${L}$. Most often ${L}$ is some deterministic function $\mathcal{L}$ of a parametrized, conformal set $\mathcal{C}(X; \lambda)$ and the label $Y$ that obeys $\lambda_1 \leq \lambda_2 \implies \mathcal{C}(X; \lambda_1) \subseteq \mathcal{C}(X; \lambda_2) \implies \mathcal{L}(\mathcal{C}(X; \lambda_1), Y) \geq \mathcal{L}(\mathcal{C}(X; \lambda_2), Y) $.

As the paper by Angelopoulos et. al. (2023) showed, conformal risk control is considerably more flexible than standard conformal prediction, while still retaining nearly identical guarantees. The work by Angelopoulos et. al. (2023) briefly touched on straightforward extensions to conformal risk control, including proving a bound on the degradation in guaranteed risk as a function of $\sum TV(Z_i, Z_{n+1})$, where $Z_i = (X_i, Y_i)$, similar to the work of Barber et. al. (2022). This paper further includes the use of data _weight_ functions (as in Barber et. al. (2022)), and shows that this can have good empirical impact on several experimental domains.

[1] Conformal Risk Control. Anastasios N. Angelopoulos, Stephen Bates, Adam Fisch, Lihua Lei, Tal Schuster. 2023.

[2] Conformal prediction beyond exchangeability. Rina Foygel Barber, Emmanuel J. Candes, Aaditya Ramdas, Ryan J. Tibshirani. 2022.

**Strengths:**

This paper is clear, and does a good job at deriving bounds for how weighted conformal risk control performs under non-exchangeability. Non-exchangeability will happen often in practice, so it is impactful to explore more robust weighting schemes and their implications. The empirical results are encouraging. It's a bit unclear as to how _useful_ the guarantees are, in the sense that they can be too loose if $\sum w_i TV(Z, Z^i)$ is very large, or more likely yet, simply unknown. When some practical knowledge about the domain is available, however, designing an appropriate weighting scheme can be effective (which is demonstrated for some of the experiments here).

**Weaknesses:**

While, again, the paper is nicely written, it is a somewhat incremental step from previous work in Barber et. al. and Angelopoulos et. al. It's also a bit of an over-claim to say that risk is _controlled_ in a non-exchangeable setting, rather what the paper does is develop a conservative upper bound for the risk under non-exchangeability that depends on quantiles that we cannot realistically know, i.e., $TV(Z_i, Z_{n+1})$.

**Questions:**

- I'm unclear if Theorem 1 holds for data-dependent weights? A similar requirement of independence is in Barber's results, and it would seem that it should be required here too. Particularly in the iterated expectation step here, I think this assumes $K \perp Z$. This is a major claim in the paper (e.g., "[...] allows weighting the data based on its statistical similarity with the test examples" in the abstract), and in the QA experiment, the weights are a function of the data points, $w_i = \textrm{sim}(X_i, X_{n+1})$.
- Otherwise, it's also unclear what the best strategy should be for selecting weighting functions (this seems rather adhoc in the experiments).
- Note that we can get the same bound in Lemma 1 directly from the analysis of Angelopoulos et. al. by defining $\tilde{g}(Z) = g(Z) - A$, which then has range $[0, B - A]$ for $g(Z) \in [A, B]$, and then it follows that
$| \mathbb{E}[g(Z)] - \mathbb{E}[g(Z')] | =  | \mathbb{E}[g(Z) - A] - \mathbb{E}[g(Z') - A] | = | \mathbb{E}[\tilde{g}(Z)] - \mathbb{E}[\tilde{g}(Z') ] |$,
and then we proceed directly with the proof in Angelopoulos et. al. to get
 $| \mathbb{E}[\tilde{g}(Z)] - \mathbb{E}[\tilde{g}(Z') ] | \leq (B-A) TV(Z, Z').$

---

> ### Author Response · Authors · 2023-11-17
>
> Thank you for your review and suggestions. We are glad that you found the paper nicely written. We address the main points below.
>
> > It's also a bit of an over-claim to say that risk is controlled in a non-exchangeable setting, rather what the paper does is develop a conservative upper bound for the risk under non-exchangeability that depends on quantiles that we cannot realistically know, i.e., TV(Z_i, Z_{n+1}).”
>
> You are right that our upper bound contains a coverage gap that is not easy to control, and we updated the manuscrit to make this more clear (see $\S3$). Nevertheless, we believe the upper bound derived in our paper can still be practically useful and it can  motivate heuristics for reducing this coverage gap via a suitable choice of the weights. We added a discussion in a new subsection ($\S3.2$), where we also provide a maxent heuristic that, among other things, justifies exponentially decaying weights in distribution shift scenarios. We hope this clarifies your concerns.
>
> > “I'm unclear if Theorem 1 holds for data-dependent weights? A similar requirement of independence is in Barber's results, and it would seem that it should be required here too. Particularly in the iterated expectation step here, I think this assumes K \perp Z. This is a major claim in the paper (e.g., "[...] allows weighting the data based on its statistical similarity with the test examples" in the abstract), and in the QA experiment, the weights are a function of the data points, w_i=sim(X_i, X_{n+1}).”
>
> Indeed, our Theorem 1 requires the independence $K \perp Z$ as in Barber et al. (2023). Our goal with the QA experiment is to assess whether the framework still works in practice when we relax this assumption (we set higher weights for questions in a “neighborhood” of $X_{n+1}$). For the other experiments ($\S4.1$ and $4.2$), we always assume data-independent weights. Note that it is possible to generalize our bound to allow data-dependent weights by conditioning the TV distance on the weights as done by Barber et al. (2023),  $\S4.5$ (paragraph “Fixed versus data-dependent weights''). While we believe that the results of $\S4.3$ are still interesting and useful in practice, we agree that the paper would benefit from more discussion about this point. We added a note in $\S3$ (last paragraph before $\S3.1$) and in $\S4.3$ to clarify this and we also updated the abstract/introduction accordingly.
>
> > “Otherwise, it's also unclear what the best strategy should be for selecting weighting functions (this seems rather adhoc in the experiments).”
>
> Although our theoretical bound on $\mathbb{E}[L(\hat{\lambda}; (X_{n+1}, Y_{n+1}))]$ holds for any choice of weights, its usefulness relies on the coverage gap in Eq. 7 being small, i.e., on our ability  to choose small weights $w_i$ for data points $Z^i$ with large total variation distance $d_{\mathrm{TV}}(Z, Z^i)$. In Barber et al. (2023), Lemma 1, it is shown that $d_\mathrm{TV}(Z, Z^i) \le 2d_\mathrm{TV}(Z_i, Z_{n+1})$, and while the true value of this term is unknown, in some cases it can be properly modeled or approximated. For instance, in the presence of distribution drift in time series, we expect  $d_{\mathrm{TV}}(Z_i, Z_{n+1})$ to decrease with $i$, therefore we should  choose weights that increase with $i$. Our experiments (e.g., $\S4.1$, Setting 3) empirically validate this assumption for the case of distribution drift. The same principle can be applied in other domains (e.g., for spatial data, one may place higher weights to points close in space to the test point).
>
> We agree  that the paper benefits from more discussion on this topic, and for that reason we updated it accordingly (please see $\S3$ after Eq. 8). In particular, we included a new subsection ($\S3.2$) that suggests a practical strategy for choosing the weights based on knowledge or assumptions about how the TV distance $d_{\mathrm{TV}}(Z, Z^i)$ varies over time, via a regularized minimization of the coverage gap using the maximum entropy principle. We also show in this subsection that this principle provides further justification for the use of weights with an exponential decay in time series under a distribution shift, which we use in our experiments of $\S4.1$ and $\S4.2$. We hope this addresses your main concerns.

---

> > ### Comment · Reviewer_wGbj · 2023-11-22
> >
> > Thanks for replying to my review comments and questions. In particular:
> >
> > - Thanks for adding the section on how to choose weights. This is helpful. Again, I'm still a little unconvinced of the utility here, as it gives us intuition about how standard CRC will fail if exchangeability does not hold, but not a concrete correction. That intuition is already quite straightforward to see from the original extensions in CRC. Still, I do appreciate the more comprehensive derivation that was done to also include weighting as in Barber (vs. the simpler bound in CRC).
> > - I'm still not sure that it's clear enough that weights need to be independent from the data points for this bound to hold (switching to "relevance" is still ambiguous). I also think that this should be stated directly as an assumption in Thm. 3.1.
> > -  Finally, though a nitpick, as per my original comments, I don't really see that fact that losses can be bounded in [A, B] vs [0, B] as a "significant" difference over CRC, as written in the related work.

---

> > > ### Author Response · Authors · 2023-11-22
> > >
> > > Thank you for your answer. We are happy that you found the new section on choosing the weights helpful. Some clarifications below:
> > >
> > > > it gives us intuition about how standard CRC will fail if exchangeability does not hold, but not a concrete correction
> > >
> > > Please note that this section not only gives the intuition about how standard CRC fails under non-exchangeability, but it also offers a *practical* solution to correct this problem, assuming estimates about the TV distance or prior knowledge about the kind of distribution shift. You can see this in the last paragraph of $\S3.2$.
> > >
> > > > I'm still not sure that it's clear enough that weights need to be independent from the data points for this bound to hold
> > >
> > > Note that we discussed this point in the last paragraph of $\S3$, just before $\S3.1$. We also talked about this in the experimental section where we use data-dependent weights ($\S4.3$; see the paragraph in purple). In any case, we followed your suggestion and added the data-independence of the weights as an assumption in Theorem 1 in our last revision.
> > >
> > > Does this alleviate your remaining concerns?

---

> ### Author Response · Authors · 2023-11-21
>
> Thank you for your time reviewing our paper. We hope you had the chance to look at our answers and updated version of the paper. Since the author response period is almost ending, we kindly request your feedback on wether we addressed your concerns. Please do let us know if further clarification is needed.

---

### Official Review · Reviewer_2JpW · 2023-11-06

**Soundness:** 4 excellent
**Presentation:** 3 good
**Contribution:** 2 fair
**Rating:** 8
**Confidence:** 4

**Summary:**

The paper describes how to perform conformal risk control for non-exchangeable data in the split conformal setting. It is shown that the proposed method has adaptive coverage guarantees, and performs well on a mixture of real-world and synthetic settings.

**Strengths:**

The paper connects two modern techniques in conformal prediction, and is thus very relevant for the community. It is well-written and easy to follow as an expert. The writing is simple, and I expect the paper will also be easy-to-follow for readers unfamiliar with conformal prediction.

**Weaknesses:**

The method combines previous work in a relatively straightforward way. The proof of Theorem 1 does not introduce new techniques. The experiments follow settings proposed in previous works. The paper is solving a completely new problem, and naturally, there are no baselines for it.

Thus, while the proposed method is novel and useful, the paper would be strengthened with a more in-depth theoretical/experimental study. Some suggestions are,
- Writing down full-conformal and cross-conformal versions of the proposed method
- Considering new experimental settings, such as established ML distribution shift datasets
- Discussion on how one can set the weights in practice
- An interpretation of Theorem 1 for a non-expert in conformal
- A discussion around the implication of Theorem 1 for specific types of distribution shift

For instance, for the synthetic experiment in Sec 4.1, some questions that can be considered are,
- What is the exact coverage guarantee of Theorem 1? Could you put it on the plot and compare it to the obtained coverage?
- Are there some "optimal" weights that give close-to optimal coverage?
Similar questions can be considered for the other experiments, although the true TV is not known, so the authors would have to think of other ways of analyzing the experiment.

## Small errors/questions:
- Just below eq. (5), C_\lambda' should be C_{\lambda'}
- Sec 2.3: "loss is nonincreasing": nonincreasing in which parameter and in what sense?

**Questions:**

Please look at questions in the "Weaknesses" section.

---

> ### Author Response · Authors · 2023-11-17
>
> Thank you for your positive reviews! We already incorporated some of your suggestions in the updated manuscript. In particular, we added some discussion following our main result in $\S3$, after Eq. 8, giving some intuition on which circumstances our results are useful; we also included a new subsection ($\S3.2$) that suggests a strategy for regularized minimization of the coverage gap inspired by the maximum entropy principle, providing further justification for the use of weights with an exponential decay in time series under a distribution shift, which we use in the experiments of $\S4.1$ and $\S4.2$.
>
> > Just below eq. (5), C_\lambda' should be C_{\lambda'}
>
> Thanks for pointing this out, we fixed the typo.
>
> > Sec 2.3: "loss is nonincreasing": nonincreasing in which parameter and in what sense?
>
> The loss should be nonincreasing with respect to $\lambda$. In practice, if we construct predictions sets of the form $\mathcal{C}\_{\lambda}(\cdot)$, with larger $\lambda$ yielding larger prediction sets, then the loss function must shrink as $\mathcal{C}\_{\lambda}(\cdot)$ grows. We clarified this in the manuscript ($\S2.3$).

---

### Meta-Review · Area_Chair_Hxsk · 2023-12-12

**Metareview:**

Conformal prediction has received a lot of attention in the ML community for its ability to provide "confidence" set on labels of given inputs without assumption on the distribution and for finite sample size. It has also be extended to control risk which can then be used to achieve some expected loss, similarly without strong assumptions. However, for both situations, one need to assume data exchangeability, which  might not hold in many application such as time series. This paper propose a natural extension of the previous analysis to non-exchangeable case.

The paper is well written and easy to follow. However, one can notice that it is quite incremental given the works (Barber etal 2022) and (Angelopoulos et. al. 2023). As many reviewers commented, the technical novelties are quite limited.

It is also a bit of an over claim to say that the risk or coverage are controlled in the non-exchangeable case. With no assumptions, anything can happen and the lower bound proposed can be vacuous. We can agree that it provides some guidance for weighting the quantile but this is purely heuristic and no longer "conformal". One could actually just learn the distribution based on the observed data and hope for the best, the current situation is not more rigorous.

In all case, the reviewer and I find the work interesting but the novelties are limited and with some over claim on what can really be controlled safely.

**Justification For Why Not Higher Score:**

The limited technical novelties given the previous works not voting above a poster presentation.

**Justification For Why Not Lower Score:**

The proposed analysis is still not trivial and the presentation is very clear and transparent.
The numerical benchmarks are interesting as well. I believe that this paper is above the acceptance threshold.

---

### Decision · Program_Chairs · 2024-01-16

Accept (poster)